# Anemia of School-Age Children in Primary Schools in Southern China Should Be Paid More Attention despite the Significant Improvement at National Level: Based on Chinese Nutrition and Health Surveillance Data (2016–2017)

**DOI:** 10.3390/nu13113705

**Published:** 2021-10-21

**Authors:** Shujuan Li, Xue Cheng, Liyun Zhao, Hongyan Ren

**Affiliations:** 1National Institute for Nutrition and Health, Chinese Center for Disease Control and Prevention, Beijing 100050, China; lisj@ninh.chinacdc.cn (S.L.); chengxue@ninh.chinacdc.cn (X.C.); 2State Key Laboratory of Resources and Environmental Information System, Institute of Geographic Sciences and Natural Resources Research, Chinese Academy of Sciences, Beijing 100101, China

**Keywords:** anemia, school-age children in primary schools, spatial disparities, influencing factors, Chinese nutrition and health surveillance, China

## Abstract

Globally, anemia among school-age children (SAC) remains a serious public health problem, impacting their growth, development, educational attainment and future learning potential. National and subnational anemia prevalence among SAC in China has not been assessed recently. Based on data from Chinese Nutrition and Health Surveillance (2016–2017), the current anemia status of SAC in primary schools in China was investigated. Anemia prevalence of SAC in primary schools in China was 4.4%, dropping 63.6% compared with that in 2002. Even though anemia was no longer a significant public health problem at the national level, there were significant spatial disparities of anemia prevalence in different areas: anemia prevalence in 63% of provinces of northern and eastern China has lowered to below 5%, while in provinces of southern China, it was still ranging from 5% to 11.0%, about 3 times of other areas. For those children in southern China, mother’s education level (OR = 1.24, *p* = 0.04) and father’s education level (OR = 1.27, *p* = 0.01) below senior high school, eating meat less than 3 times per week (OR = 1.18, *p* = 0.01) were risk factors of anemia. Older age (OR = 0.35–0.72, *p* < 0.01) was a protective factor. Targeted interventions should be taken to improve SAC anemia in Southern China, despite anemia of SAC in primary schools no longer being a significant public health problem.

## 1. Introduction

Anemia is a global public health problem affecting 27 percent of the world’s population [1]. Children and women are the most vulnerable population to anemia. According to a WHO report, the global anemia prevalence of school-age children (SAC) was 25.4%, following preschool children under 5 years old (47.4%) and women of childbearing age (30.2%) [2]. SAC are in the critical stage of growth and development, and anemia may lead to muscle function and work ability reduction [3], cognitive function impairment and academic performance reduction [4], and may even affect future performance. There are already a lot of studies on anemia of pre-school children [5,6] and women of childbearing age [5,7]; however, there are relatively fewer studies on SAC anemia [8], which should be paid more attention to.

Anemia has multifactorial causes including genetic mutations in hemoglobin genes, acute and chronic blood loss, inadequate nutritional intake, infectious diseases and other factors [1,7]. It is known that some key demographic and socioeconomic factors influence SAC anemia status, including gender [8,9], family wealth [9], parents’ education [10], nutrition intake [11,12] and nutrition status [13,14], and area of residence [9,11,13]. The influencing factors directly or indirectly affect children’s nutrition and medical conditions, and eventually affect anemia status. Iron deficiency is the dominant cause (≥60%) of anemia globally [1], and in China, it is more than 90% [15]. In China, iron deficiency is closely related to nutrition status. Due to the development of the social economy and the improvement of residents’ living standards, children’s anemia has been greatly improved in China: anemia of children aged 6 to 11 years in China has decreased from 12.1% in 2002 to 5.0% in 2013 [16]. However, the imbalanced development between urban and rural areas in China has led to significant differences in nutrition and anemia status among SAC. According to Luo et al., the anemia prevalence of primary school students in Qinghai and Ningxia poor rural counties was 24.9% in 2009, which was twice the average anemia prevalence (12.1%) in China [10]. Shen et al.’s study showed that anemia prevalence of primary school students in the poverty-stricken counties of Guangxi and Yunnan was 20.5% in 2011 [17].

There were some studies that discussed the spatial disparity of SAC anemia. Wang et al.’s study demonstrated that the lower class of rural had the highest anemia prevalence (12.0%) while the large coastal city had the lowest anemia prevalence (5.2%) [18]. Luo et al.’s study showed that anemia prevalence of Han students aged 7–14 in Hainan province was 24.1%, and was 19.6% in Gansu province, while it was 2.0% in Beijing [19]. Although those two studies discussed the SAC (7, 9, 11 and 14 years old) anemia at different geographical groups and at provincial levels separately, there was no specific study on the spatial disparity of SAC anemia prevalence at continuous age. Exploring the spatial disparity and its influencing factors can provide a solid foundation for anemia prevention.

To investigate the current anemia status of SAC in primary schools, we used data from Chinese Nutrition and Health Surveillance (CNHS, 2015–2017) to evaluate the hemoglobin level and anemia status, and explore regional disparities and influencing factors (including demographic factors, socioeconomic factors, family status factors, and regional factors), and offer suggestions of targeted intervention strategies.

## 2. Materials and Methods

In this study, data of SAC in primary schools from CNHS (2015–2017) were used. CNHS (2015–2017) is a national surveillance representative at both the national and provincial levels covering 31 provinces of China (not including children from Taiwan, Hong Kong and Macao). The children’s survey was implemented in 2016–2017.

**Sampling method.** Multistage stratified cluster sampling was used to monitor the nutritional health of children from 0 to 17 years in China. According to the principle of representative, 275 monitoring points were chosen from all the administration counties/county-level cities/districts which were categorized into 4 strata (big cities: BC, small-middle sized cities: SMC, ordinary rural counties: OC, and poverty-stricken rural counties: PC, based on the population size and the definition of urban or rural from National Bureau of Statistics of the People’s Republic of China in China) [20]. At least 28 children in each grade (1–6 grades in primary school) were surveyed at each monitoring point. After signing the informed consent form, the selected children were confirmed as the respondents. 263 monitoring points were actually completed. Due to the small actual sample size in Tibet, we did not calculate provincial children’s anemia prevalence in Tibet.

**Survey method.** The children survey included an inquiry survey (household and individual questions), anthropometric measurements (weight, height, waist circumference, and blood pressure), laboratory test (hemoglobin, ferritin, transferrin receptor, high sensitive C-reactive protein, blood glucose and so on) and dietary survey (3 consecutive days’ food weight record, 3 consecutive days’ 24-h dietary recall, and Food Frequency Questionnaire) [20]. Food Frequency Questionnaire (FFQ) results were used for evaluating the dietary status of SAC since every child took this survey and only 8 of 28 children took 3 consecutive days’ food weight record. FFQ inquiry was answered by the parents for SAC in primary schools. The Fasting venous blood samples were collected and detected by the Hemocue method (Hemocue 201+ hemoglobin detect meter). Due to the data availability, we did not use the ferritin-related data.

**Sample size description.** 39,469 SAC aged 6–11 years with hemoglobin data were included in this study, including 19,647 (49.8%) boys and 19,822 (50.2%) girls; 18,649 (47.2%) living in cities and 20,820 (52.8%) living in rural areas. After eliminating those children who did not have personal basic information, 38,986 children were included in the univariate logistic analysis of influencing factors to screen out the relevant factors. For those children distributed in the southern region with anemia prevalence higher than 5%, a multivariate logistic regression analysis was used to detect the risk influencing factors. The study procedure was shown in Figure 1.

**Anemia Definition.** For children aged 6 to 11 years old, with altitude <1000 m, hemoglobin value <115 g/L is defined as anemia, hemoglobin value ≥110 g/L and <115 g/L as mild anemia, hemoglobin value ≥80 g/L and <110 g/L as moderate anemia, hemoglobin value <80 g/L as severe anemia [21]. For areas with an altitude of more than 1000 m, adjustments were made to the measured hemoglobin concentration according to WHO standards [21].

**Influencing factor variables.** The selection of variables included in the influencing factor analysis was specified based on a review of the published literature pertaining to predictors/risk factors of anemia among school-aged children. These factors included demographic factors [8,9], socioeconomic factors [9,10], family and personal status factors [12], and regional factors [9,18] (Figure 1). We considered the following factors: gender (boy vs. girl), age (years), urban/rural, city type (BC, referred to the central urban areas with a population of more than 1 million municipalities directly under the central government, cities under separate planning and provincial capital cities; SMC, referred to all the districts and county-level cities outside the central urban areas of the above-mentioned big cities and the county-level cities or districts among the 592 poverty-stricken counties; PC, county-level cities or districts are removed from the 592 counties identified in the “2001–2010 national program for poverty alleviation and development in rural areas”; OC, counties except PC) [22], Parents’ education level (junior middle school and below vs. senior high school/technical secondary school and above),Toilet type (sanitary toilet vs. non-sanitary toilet, sanitary toilet includes water flushing toilet and sanitary dry toilet), Water source (safe water source vs. unsafe water source, safe water source included tap water after centralized purification, water from protected well or spring and bottled water; unsafe water source included water from unprotected well and spring water, rivers and lakes ditch pond water, collected rain, snow and other source), Resident students (yes vs. no), meat intake (≥3 times per week vs. <3 times/week), regional divisions (the southern part vs. the other part).

**Statistical analyses.** SAS 9.4 software (SAS Institute Inc., Cary, NC, USA) was used to process and analyze the data to obtain the anemia status and their influencing factors. Anemia prevalence results were weighted based on the 2010 Population Census of the People’s Republic of China data (considering age, gender and urban/rural). The influencing factors of anemia were analyzed by univariate logistic regression to screen out the more relevant influencing factors. Those more relevant influencing factors were included in the multivariate logistic regression model. For those children in high prevalence regions, multivariate logistic regression was used to analyze the risk factors. For univariate logistic regression and multivariate logistic regression models, the dependent variable was present vs. absent anemia, and the independent variables were the influencing factors we considered. A stepwise method of variables entry was used. The alpha significance level was set at 0.05.

**Spatial analysis method.** ArcGIS 10.7 software was used to map monitoring point’s level and province level children anemia prevalence.

**Ethical approval.** This study was approved by the ethics review committee of the National Institute for Nutrition and Health, Chinese Center for Disease Control and Prevention (no.201614).

## 3. Results

### 3.1. Hemoglobin and Anemia Status

The hemoglobin concentrations of SAC in primary schools were 133.2 ± 10.8 g/L. The hemoglobin concentrations were 133.5 ± 10.9 g/L, 132.8 ± 10.8 g/L, 134.0 ± 10.5 g/L, and 132.4 ± 11.1 g/L for boys, girls, children living in urban areas and children living in rural areas. There were significant differences between urban and rural areas and between boys and girls (*p* < 0.01). As children grow up, the hemoglobin concentration increased accordingly (Table 1).

Anemia prevalence of Chinese SAC in primary schools was 4.4%, and boys’ was 4.2% and girls’ was 4.5% (Table 1). 98.9% of the anemia were mild and moderate anemia (53.5% were mild anemia, 45.3% were moderate anemia, 1.2% were severe anemia). With age increased, anemia prevalence showed a downward trend. There were significant differences in children’s anemia between urban and rural areas (*p* < 0.01). Anemia prevalence of urban children was 3.5% and of rural children was 5.0%.

### 3.2. Regional Disparity of Anemia

Anemia prevalence of SAC in primary schools in China showed significant regional disparities (Figure 2a–c). Children anemia prevalence in most provinces of China has fallen below 5%, reaching the normal level defined by WHO [21] (a site with an anemia detection rate of ≥40% was defined as having a severe epidemic, a site with a rate of 20.0–39.9% as having a moderate epidemic, a site with a rate of 5.0–19.9% as having a mild epidemic and a site with a rate of ≤4.9% as normal). In other provinces, anemia prevalence was at a mild epidemic level, ranging from 5% to 11.0%, among which Chongqing (11.0%) and Hainan (10.8%) had anemia prevalence higher than 10%. Boys with high anemia prevalence were mainly concentrated in southern China, and Chongqing had the highest anemia prevalence (12.3%); girls with high anemia prevalence were mainly concentrated in southern China and Qinghai, and Gansu province, with Hainan (11.9%) having the highest anemia prevalence.

Children’s anemia at 263 survey points presented different prevalence levels (Figure 3). 70.3% of the surveillance points (184/263) had anemia prevalence below 4.9%, widely distributed all over China. 29.0% (76/263) surveillance points had mild anemia problems, and 3 points (Jiangjin District of Chongqing, Guangnan county and Jingdong Yi Autonomous County of Yunnan) had moderate anemia problems. In total, the anemia prevalence of southern areas was about 3 times of the other area.

### 3.3. Influencing Factors of SAC Anemia

According to univariate analysis (Appendix A Table A1), being a girl, being a resident student, grandparents as the first caregiver, parents working outside, drinking unsafe water sources, accessing a non-sanitary toilet, parents with lower education levels and eating meat less than 3 times per week were risk factors of children anemia (*p* < 0.05). Older age, was protective factors of anemia (*p* < 0.05). As to Southern China, the risk factors included grandparents as the first caregiver, and parents working outside, having difficulty accessing safe drinking water, parents with lower education levels, and eating meat less than 3 times per week (*p* < 0.05). The protective factors included older age (*p* < 0.05).

Table 2 showed the factors associated with the anemia of SAC in primary schools derived from multivariate logistic regression analyses. For children in the whole country, an older age (OR = 0.37–0.82, *p* < 0.05) was associated with lower odds of developing anemia. In contrast, being a girl (OR = 1.14, *p* = 0.01), living in ordinary rural county (OR = 1.39, *p* < 0.01), living in Southern part of China (OR = 2.83, *p* < 0.01), lower education level of mother (OR = 1.23, *p* = 0.01) and father (OR = 1.18, *p* = 0.04), eating meat less than 3 times per week (OR = 1.12, *p* = 0.04) were associated with higher odds of developing anemia.

For children in Southern area of China, growing older (8, 9, 10, 11 years old) (OR = 0.35–0.72, *p* < 0.01) was associated with lower odd ratio of anemia. Lower education level of mother (OR = 1.24, *p* = 0.04) and father (OR = 1.27, *p* = 0.01), eating meat less than 3 times per week (OR = 1.18, *p* = 0.01) were associated with higher odd ratio of anemia.

## 4. Discussion

In this study, we explored anemia prevalence of SAC in primary schools based on national and provincial representative data of CNHS (2015–2017) and analyzed the influencing factors of anemia, which could provide important information for SAC nutrition intervention strategy.

We found that the anemia prevalence of SAC in primary schools in China was 4.4%, which illustrated anemia of 6–11 years aged children was no longer a significant public health issue at the national level in China [21]. Compared with the anemia prevalence of SAC in developing countries such as Mexico (children 5–14.99 years old, 12.0% [23]), India (Odisha: children aged 6–12 years old, 68.9% [24]), and low-income and high infection countries such as Ethiopia (eastern: children aged 5–15 years old, 27.1% [25]; southwest: children aged 6–14 years old, 37.6% [26]), SAC anemia in China was in a better state. Compared with the US SAC anemia state (girls aged 12.00–14.99 years: 4.0%) [23], China SAC anemia prevalence was slightly higher. At the same time, China is the most populous country in the world, with the largest or one of the largest populations of SAC (104.3 million School-age Children in Primary Schools in 2020 according to the Ministry of Education of the People’s Republic of China [27]). So even if the burden of anemia may be lower than some other upper-middle-income countries, the absolute number of SAC suffering from anemia in China is likely the highest in this World Bank income classification group, which should be paid more attention.

Anemia prevalence of children aged 6–11 years old in China has decreased from 5.0% in 2012 [28] and 12.1% in 2002 [28]. One possible reason for this improvement might be nutrition improvement brought by the economic development and improvement of people’s living level [9,29]. According to the results of CNHS (2010–2013), the intake of livestock and poultry in China reached 89.7 g in 2012, increased by 11.1 g compared with 2002, and the intake in rural areas increased from 68.7 g in 2002 to 81.2 g in 2012 [28], which might contribute to the improvement of SAC anemia. The results also reflected great improvements in SAC nutrition status. Nutrition Improvement Program for Rural Students of Compulsory Education issued by the State Council in October 2011 [30] contributed a lot to the anemia improvement in primary students in rural areas [31,32,33]. On the other side, due to the great population size in China, the burden of children’s anemia is still heavy. According to the disease burden of iron-deficiency anemia (IDA) among children and adolescents in China, disability-adjusted life year (DALY) of IDA was 445.64~1080.31 person-year per 100,000 and in 16 provinces it exceeded 800 person-year per 100,000 [34]. Moreover, anemia status is closely related to food iron intake and tends to have relatively high volatility in a vulnerable area and vulnerable population. Despite the great improvement in children’s anemia, it should not be neglected.

While the anemia prevalence of children in eastern and northern provinces of China had dropped to the normal range defined by WHO [21], anemia prevalence in some provinces of southern China was still high, ranging from 5% to 11.0%, which was about 3 times of other areas. In other studies, spatial disparities were found among the lower class of rural areas and the large coastal cities [18], east developed region and west less-developed regions [19]. In this study, children in the southern area of China were found more suffering from anemia. The possible reason might be those areas were concentrated in poverty-stricken rural counties. For children in Hainan, the higher anemia prevalence might relate to parasitic infections such as soil-transmitted nematodes [19]; for children in Chongqing, the specific reasons for anemia should be explored.

The occurrence of anemia in children is affected by many factors, including ecology, climate, geography, social and economic factors, such as family income, parents’ education level, and biological factors [7]. In this study, demographic factors, socioeconomic factors, family status factors, and regional factors were analyzed. Gender, age, meat intake frequency, living areas, parents’ education level, whether parents going out for work, main caregiver, source of drinking water, type of family toilet, and spatial disparity were found as significant influencing factors of childhood anemia. Economic growth is closely related to anemia reduction. Since with the improvement of social and economic status, there will be easier access to abundant food, especially meat, and better living conditions, safer water supply, better sanitation and more convenient transportation [18]. Moreover, these people usually have higher education levels, better economic status and employment opportunities, which will affect the nutritional and hygienic benefits of children and improve anemia status eventually. According to the global burden of anemia, 60% of anemia was IDA [1], which was related to nutrition status especially animal source foods [5]. In this study, eating meat more than 3 times per week significantly reduced the risk of anemia, which was consistent with one other study in China [12], since animal products intake would ideally address nutritional anemia [5]. Parents’ higher educational level was the protective factor to anemia in this study, which was consistent with many other studies [10,35]. Parents with higher education levels tend to have higher income and more nutrition and health knowledge, which eventually affects children’s nutrition intake and anemia. In total, parents’ education, sources of drinking water, toilet types and regional differences between urban and rural areas were indicators reflecting the economic and social status, which was consistent with the results of Luo [29] and other research results [9,18].

For children in southern China vulnerable areas, and those in lower social and economic status, targeted prevention strategies should be taken to improve anemia. Specific nutritional policies like the Nutrition Improvement Program for Rural Compulsory Education Students should pay attention to the intake of animal food of students, especially for younger students and girls. Nutrition education should be strengthened in schools and make sure SAC realize the importance of dietary diversity and implement it in daily life.

The strengths of this study lie in its large representative sample of Chinese primary school-aged children. The CNHS (2015–2017) was national surveillance and was implemented by the Chinese Centre for Disease Control and Prevention. All surveillance sites strictly followed the standardized protocols and data collection procedures. In addition, we collected information on a wide range of potential associated factors.

Despite large national and provincial representative samples, there were some limitations in this study. Firstly, this study did not include serum ferritin results, so the anemia type could not be defined. Although most anemia in China was iron deficiency anemia according to the previous studies [5,36], further study should be conducted to explore this aspect. Secondly, further study should be carried out to explore the relationship between dietary nutrition and anemia status, which will provide effective strategies for children anemia improvement.

## 5. Conclusions

Anemia of SAC in primary schools is no longer a public health issue in China since anemia prevalence has dropped below 5%. However, there are significant regional disparities in anemia. While SAC anemia in 63% of provinces of China has reached the normal level defined by WHO, in a few provinces of southern China, anemia prevalence is still at the mild epidemic level, ranging from 5% to 11.0%. For children in regions with higher anemia prevalence, growing older was a protective factor for anemia, and parents’ education level below senior high school, and eating meat less than 3 times per week were risk factors for children’s anemia. A targeted intervention strategy should be taken to improve anemia in those vulnerable areas.

## Figures and Tables

**Figure 1 nutrients-13-03705-f001:**
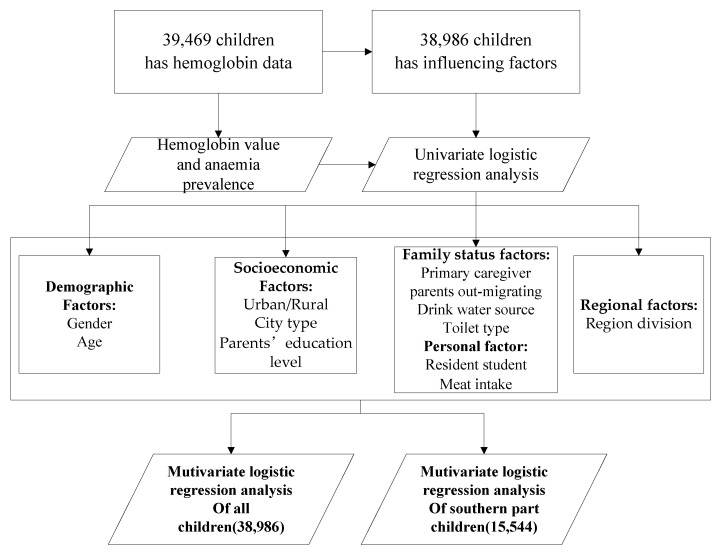
Flowchart of study procedures.

**Figure 2 nutrients-13-03705-f002:**
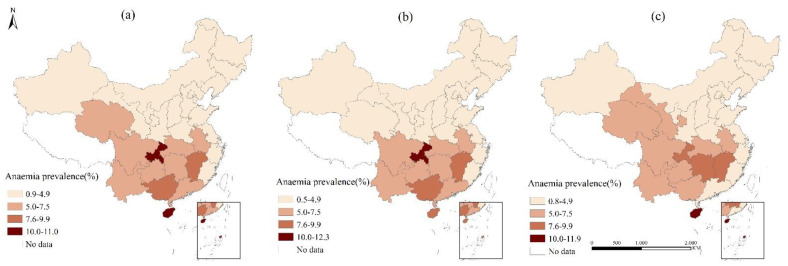
(**a**) Anemia prevalence of children; (**b**) anemia prevalence of boys; (**c**) anemia prevalence of girls.

**Figure 3 nutrients-13-03705-f003:**
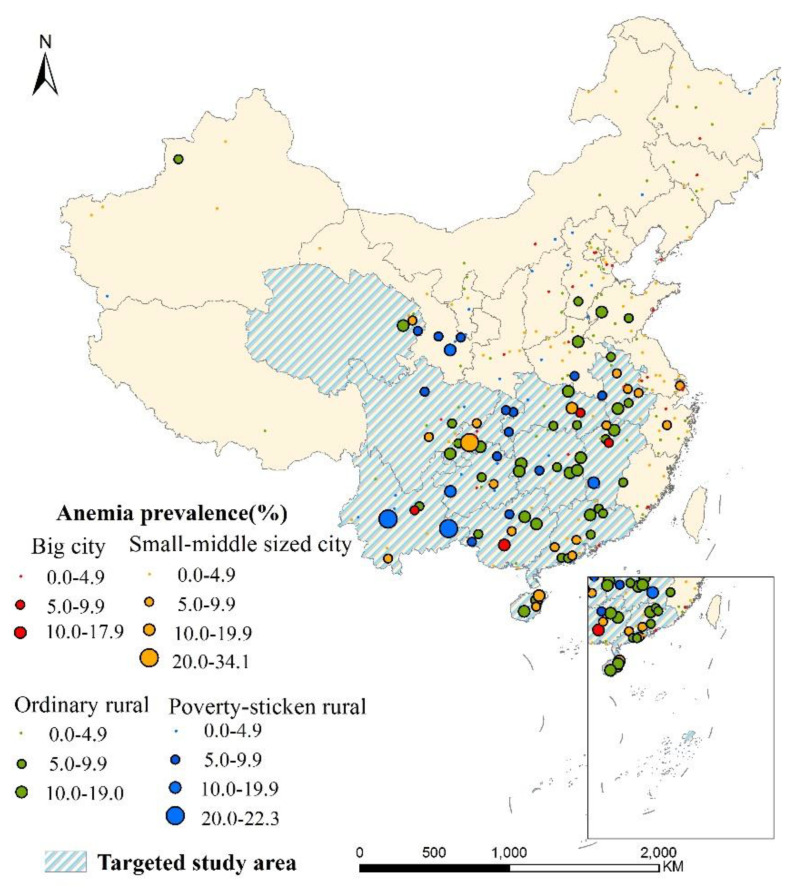
Anemia prevalence of SAC in primary schools at surveillance points.

**Table 1 nutrients-13-03705-t001:** Hemoglobin and Anemia status of SAC in primary schools.

Characteristic	N(%)	Hb(g/L)	Anemia Prevalence	
		X ± SD	%	95%CI
Total	39,469(100%)	133.18 ± 10.84	4.37	3.22–5.52
Boy	19,647(49.8%)	133.52 ± 10.90 *	4.24 *	2.94–5.55
Girl	19,822(50.2%)	132.84 ± 10.77 *	4.52 *	3.44–5.59
Child’s age (in years)				
6	3937(10.0%)	129.51 ± 10.73	5.71	4.20–7.22
7	6916(17.5%)	130.75 ± 10.63	5.33	4.02–6.65
8	7250(18.4%)	132.49 ± 10.82	4.35	3.46–5.24
9	7240(18.3%)	133.60 ± 10.31	4.21	2.71–5.70
10	7004(17.7%)	134.65 ± 10.56	4.01	1.62–6.39
11	7122(18.0%)	136.40 ± 10.71	3.25	1.48–5.01
Urban/Rural				
Urban	18,649(47.2%)	134.03 ± 10.51 *	3.50 *	1.13–5.87
Rural	20,820(52.8%)	132.42 ± 11.08 *	4.99 *	3.96–6.02

* *p* < 0.01.

**Table 2 nutrients-13-03705-t002:** Factors associated with children anemia based on multivariate logistic regression analyses.

Influencing Factors	Reference	All Children		Children in Southern Area	
		OR(95%CI)	*p*	OR(95%CI)	*p*
Gender					
Girl	Boy	1.14(1.03–1.26)	0.01	NS	NS
Age					
7	6	0.82(0.69–0.96)	0.02	0.95(0.77–1.17)	0.63
8	6	0.67(0.56–0.79)	<0.01	0.72(0.58–0.89)	<0.01
9	6	0.49(0.41–0.59)	<0.01	0.58(0.46–0.72)	<0.01
10	6	0.43(0.36–0.52)	<0.01	0.44(0.35–0.56)	<0.01
11	6	0.37(0.30–0.45)	<0.01	0.35(0.27–0.45)	<0.01
City Type					
SMC	BC	1.09(0.88–1.34)	0.42	NS	NS
OC	BC	1.39(1.12–1.71)	<0.01	NS	NS
PC	BC	1.25(1.00–1.57)	0.05	NS	NS
Zone					
South area	The other area	2.83(2.54–3.16)	<0.01	NA	NA
Mother’s education					
Low	High	1.23(1.04–1.45)	0.01	1.24(1.01–1.52)	0.04
Father’s education					
Low	High	1.18(1.00–1.37)	0.04	1.27(1.05–1.54)	0.01
Meat type					
<3 times/week	≥3 times/week	1.12(1.00–1.25)	0.04	1.18(1.04–1.35)	0.01

A stepwise method of variables entry was used. NS, the variable was not entered into the regression model. NA, not available.

## Data Availability

The data is not allowed to disclose according to the National Institute for Nutrition and Health, Chinese Center for Disease Control and Prevention.

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
