# Peer review of "Anemia of School-Age Children in Primary Schools in Southern China Should Be Paid More Attention despite the Significant Improvement at National Level: Based on Chinese Nutrition and Health Surveillance Data (2016–2017)"

_nutrients, 2021, doi:10.3390/nu13113705_

Round 1

Reviewer 1 Report

The work by Li and colleagues assesses the prevalence of anemia among school-aged children (SAC) in China, using the most recent Chinese Nutrition and Health Surveillance data. As the authors rightly point out anemia in SAC is not studied as extensively as it is among pre-school children and women of reproductive age. As a result, this paper has the potential to contribute significantly to the literature. However, significant revisions are required.

Overall comments: The article is not well written, with several grammatical inconsistencies, which make reading the article difficult. For e.g. line 1-2 of the Abstract can be phrased as, “Globally, anemia among school-aged children (SAC) remains a serious public health problem, impacting their growth, development, educational attainment and future earning potential. National and subnational anemia prevalence among SAC in China has not been assessed recently.” The authors should highlight that China is the most populous country in the world, with the largest or one of the largest populations of SAC (if possible, provide the actual number) so even if the burden of disease may be lower than other upper middle income countries, the absolute number of SAC suffering from anemia is likely the highest in this World Bank income classification group.

For reporting numerical results in text, only report up to one decimal place. Specific comments: Lines 39-41: Provide references Lines 88 – 90: Provide percentages along with absolute numbers of sample size Lines 108 – 110: Provide references for studies used to identify risk factors for anemia among SAC Lines 118: Why was household wealth not considered in the statistical models? Intake of other iron-rich foods e.g. green leafy vegetables, fortified grains/cereals, supplements etc.? Lines 163: Define survey points. Lines 206 – 208: India and Ethiopiaare low-income countries and are much less socially developed than China. The prevalence in China should be compared with countries that are socioeconomically similar to China. Table 1: Include percentage as well.

Reviewer 2 Report

The Manuscript is sound and the materials are well presented. However, I have included some comments (see attached).

Lines 88 to 90: It would be preferable to use percentages here rather than numbers. To ease the reader's work   (from lines 88 to 90). 

Lines 123 to 124 :

The use of multivariable logistic regression model.

I wonder WHY was multilevel multivariable logistic regression not used? given the hierarchical nature of data (multistage dataset). The estimates and/or SE computed using multivariable logistic regression model will be biased (over/underestimated). therefore some insignificant factors may seem significant when they are not and vice versa.

In addition, there are many other factors that might influence children's ODD/risk of anaemia such as dietary diversity (other nutritional factors) as highlighted in the study limitation, infection but not included in the model. The multivariable analysis would not take into account these unobserved factors, but would be accounted for in the multilevel multivariable regression model.

The use of multilevel multivariable logistic regression model is warranted here.

Table 1: 

Logically, the observed mean is reported with standard deviation (SD). but not with the standard error. The standard error is reported with an estimated mean from regression for example.

Please update there and replace the SE with SD

Reviewer 3 Report

This study describes the prevalence of anemia in school-age children in China. In addition to the description of the phenomenon, the demographic, socioeconomic, family status, and geographical factors that influence the prevalence of anemia are analyzed and discussed.

General comment: This study focused on a large representative sample of the Chinese population; it is of value and interest to a broad audience. However, the theoretical framework provided is too synthetic. Authors should submit a richer literature review, which helps the reader understand the state of the art, mainly why they chose to focus their work on certain socioeconomic and geographical variables. In addition, the authors should describe the analyzes and study results in a more precise and detailed way because the description of the study at this time may be unclear.

Introduction – line 37: When the reference to a study is reported in square brackets in the text, sometimes the authors leave a space between the preceding word and the bracket, other times, they do not. They should correct this so that it is consistent throughout the text.

Introduction – line 39: The authors state that extensive literature on anemia of preschool children and women of childbearing age exists. They should name at least two of the most relevant studies for each population. Similarly, they claim that few studies exist on anemia of school-age children. They should cite the most significant studies.

Introduction – line 42: The authors declared that “anemia has multifactorial causes involving a complex interaction between nutrition, infectious diseases, and other factors.” The authors should articulate in more detail the causes that contribute to determining a diagnosis of anemia and possibly mention which demographic, socioeconomic, and environmental factors are associated with a higher risk of anemia. A deeper reflection on these aspects is essential to make the reader understand the study's rationale and the focus on the independent variables treated in the analyzes.

Introduction – line 50: When the authors cited a study with more than one author (for example, Luo's study), they should cite as follows: First author et al [number of the reference], or First Author and collaborators [number of the reference]. The authors should correct this throughout the text.

Introduction – line 52/53: I believe there is an error in this sentence; the authors should clarify that the study of Shen et al focuses on the prevalence of anemia.

Introduction – line 66: The authors should introduce what “influencing factors” they consider.

Materials and Methods – line 87: What questions and medical and laboratory tests are included in the study? For example, does the “dietary survey” only include a question related to meat intake? Who responded to the survey? Children or their caregivers? Has other clinical information been evaluated besides the hemoglobin values that could be useful in framing the sample? In this paragraph, the authors should describe the method in detail.

Materials and Methods – line 87: Could you please include the percentages related to the sample size description?

Materials and Methods – line 87: I do not understand why the first logistic regression is univariate and the next one is multivariate. What is the difference between the two? A detailed description of the dependent and independent variables would help the reader understand which analytic strategies were used.

Materials and Methods – line 96: I suggest rephrasing the classification of anemia more concisely. For example, the authors could write that hemoglobin value ≥110 g/L and <115 g/L defined mild anemia.

Materials and Methods – line 100: I suggest describing the adjustments concisely in a table without reporting them extensively in the text.

Materials and Methods – line 108: The authors declared that the “selection of variables included in the influencing factor analysis was specified based on a review of the published literature." They should cite the studies they relied on upon in defining the variables. These studies should be described briefly in the introduction.

Materials and Methods – line 110: I suggest defining precisely (as done for toilet type, water source, and parental education) the variables socioeconomic, family status, and regional factor.

Materials and Methods – line 119: The authors should explicitly describe the dependent variables and the independent variables of the three logistic regressions performed. It is currently unclear and challenging to understand which variables contribute significantly to determining the prevalence of anemia and which do not. For example, in the results section (line 140-141), the percentage of participants characterized by mild, moderate, and severe anemia is defined, but it is not clear how this variable (severity of anemia) was included in the analyzes. Is it treated only as a descriptive variable? Regarding the logistic regressions, is the dependent variable the prevalence of anemia, classified as a dichotomous variable (present vs. absent anemia - binomial logistic regression), or as a 3-level variable (mild, moderate, severe anemia - multinomial logistic regression)? Authors should describe in detail the analyzes conducted to help the reader understand the results as a whole.

Results – line 137: I suggest always using the same verbal label to define the gender of the participants. In the manuscript, reference is made to boys/girls (line 135) and subsequently to men/women (line 137) or females (line 183).

Results – line 140: The prevalence of anemia in boys and girls is shown in Table 1, not Table 3

Results – line 145: I suggest inserting in addition to N also the respective percentages in brackets N (%)

Results – line 196: I suggest using the word “gender” and not “sex" in a way that is consistent with what is done in the text

Discussion – line 240: I suggest replacing "sex" with "gender."

Discussion – line 241: I suggest replacing "guardian" with “caregiver."

Discussion – line 246: I suggest rephrasing the following sentence: "Moreover, these people usually have higher education level, economic status and employment opportunities, which will improve anemia status eventually.” It should be clear that education and economic status do not directly affect anemia but rather an effect mediated by the nutritional and hygienic benefits of higher education and better economic status.

Discussion – line 263: I suggest rephrasing the following sentence: “… dietary diversity should be aware of and implemented in daily life.”

Appendix – Table 1: The variable “soil iron level” is not introduced and described in the text. Authors should briefly define why this variable is important and cite at least one relevant study. Furthermore, in the Materials and Methods section, it is necessary to describe all the included variables in the analyses. At the moment, there is no reference to this variable in the text.
